# Unified K-Means Clustering with Label-Guided Manifold Learning

**Qianqian Wang** [1]  **Mengping Jiang** [1]  **Zhengming Ding** [2]  **Quanxue Gao** [1]

## Abstract

K-Means clustering is a classical and effective unsupervised learning method attributed to its simplicity and efficiency. However, it faces notable challenges, including sensitivity to random initial centroid selection, a limited ability to discover the intrinsic manifold structures within nonlinear datasets, and difficulty in achieving balanced clustering in practical scenarios. To overcome these weaknesses, we introduce a novel framework for K-Means that leverages manifold learning. This approach eliminates the need for centroid calculation and utilizes a cluster indicator matrix to align the manifold structures, thereby enhancing clustering accuracy. Beyond the traditional Euclidean distance, our model incorporates Gaussian kernel distance, K-nearest neighbor distance, and low-pass filtering distance to effectively manage data that is not linearly separable. Furthermore, we introduce a balanced regularizer to achieve balanced clustering results. The detailed experimental results demonstrate the efficacy of our proposed methodology.

## 1. Introduction

Clustering is an unsupervised learning technique aiming at organizing data groups that exhibit similar characteristics without labels (Nie et al., 2023). It has been widely applied in Web data analysis and information retrieval (Fu et al., 2023; He et al., 2024). K-Means (Hartigan & Wong, 1979; Lloyd, 1982) is one of the simplest and most popular clustering algorithms (Liu et al., 2023). Its core concept is partitioning data points into $c$ mutually exclusive clusters, and with each centroid representing a cluster, each data point is assigned to the cluster associated with its nearest centroid. The algorithm achieves this by minimizing the

sum of squared distances between each data point and its corresponding centroid within a cluster.

While K-Means excels in processing linearly separable data, it has limitations when dealing with nonlinearly separable data (Bai & Liang, 2020; Lu et al., 2024; Cheng et al., 2024). This is because it utilizes Euclidean distance to measure the distance between data points, making it incapable of dividing nonlinearly separable clusters. To address these limitations, researchers have proposed various improvement methods. For instance, Kernel K-Means (KKM) (Schölkopf et al., 1998) introduces a kernel function to map data into a high-dimensional space, where the transformed data become linearly separable. This has led to the development of the Multi-Kernel K-Means (MKKM) algorithm (Du et al., 2015; Yao et al., 2020; Wang et al., 2022; Liu, 2022), which utilizes multiple kernel functions to capture the nonlinear structure of data better. Another solution is spectral clustering (Ng et al., 2001; Zhou et al., 2020; Ding et al., 2024), an algorithm based on graph theory (Yang et al., 2023b), which utilizes the similarity matrix between data points to project them into a low-dimensional feature space and then applies the traditional K-Means algorithm, making it suitable for complex nonlinear data and irregularly shaped clusters (Fettal et al., 2023; Yang et al., 2023a).

However, the aforementioned K-Means methods all require the initialization of cluster centroids, which can easily lead to suboptimal local solutions rather than the optimal global solution (Nie et al., 2022). K-Means++ (Arthur & Vassilvitskii, 2006) adopts an improved initialization method that selects initial centroids according to a probabilistic distribution, resulting in a more uniform distribution of initial centroids and reducing the risk of falling into inferior local minima. Nie *et al.* proposed the Coordinate Descent Method for solving K-Means (CDKM), which can solve a better local minima of the objective function and effectively address the problem of forming empty clusters (Nie et al., 2022). Nevertheless, these methods still require the computation of cluster centroids, which are calculated based on the mean of all points within a cluster. Consequently, outliers can significantly impact the clustering results (Wang & Su, 2011; Huang et al., 2021; Heidari et al., 2024). Both (Lu et al., 2023) and (Pei et al., 2023) introduce centerless variants of K-Means, which minimize the distance between samples within the cluster instead of the distance between samples

[1]School of Communication Engineering, Xidian University, Xi'an, China. [2]Department of Computer Science, Tulane University, New Orleans, LA. Correspondence to: Quanxue Gao <qx-gao@xidian.edu.cn>.

*Proceedings of the 42$^{nd}$ International Conference on Machine Learning*, Vancouver, Canada. PMLR 267, 2025. Copyright 2025 by the author(s).

and the centroids. Therefore, these methods eliminate the need for initializing and updating the cluster centroid, thus preventing their negative influence and leading to more robust results and improved clustering performance.

Although these methods address the centroid initialization problem, they generally overlook the issue of balanced clustering, resulting in a bias towards larger clusters and smaller clusters being absorbed by larger ones (Malinen & Fränti, 2014). Even worse, all samples might be assigned to a single cluster. Balanced clustering, on the contrary, seeks to minimize the mean squared error (MSE) meanwhile considering the balance of cluster sizes, thus avoiding trivial solutions. Regularized K-Means (RKM) (Lin et al., 2019) introduces regularization terms to control the size of each cluster for achieving balanced clustering, while Discrete and Balanced Spectral Clustering with Scalability (DBSC) modifies the objective function of spectral clustering to promote balanced spectral clustering (Wang et al., 2023). However, these methods ignore the manifold structure and thus cannot exploit explicit cluster distribution.

To mitigate these problems, we propose a novel unified K-Means clustering framework. We transform traditional K-Means into a label-guided manifold learning version without the need to compute the cluster centroids, thereby improving the robustness of clustering. Specifically, we use the learned cluster indicator matrix to construct a similarity matrix, which ensures the consistency of the data manifold structure and cluster labels. Moreover, we maximize the balanced regularization term to ensure the balance of clustering results and provide the theoretical proof of its effectiveness. Finally, we further introduce four different distances, *i.e.*, Euclidean distance, K-nearest neighbor distance, Gaussian kernel distance, and low-pass filtering distance, which are suitable for nonlinearly separable data and make it adaptive to datasets of various structures. The main contributions of this work can be summarized as follows:

- **Decentralization:** We propose a unified K-Means clustering framework that eliminates centroid calculations and hence improves the robustness to outliers.

- **Label-Guided Manifold Learning:** We construct the similarity matrix using a learnable cluster indicator matrix, ensuring the consistency of the data's manifold structure and its labels.

- **Cluster Balance:** We introduce a balanced regularization term that maximizes $\ell_{2,1}$-norm of the cluster indicator matrix to ensure the balance of clustering results.

- **Four Distance Metrics:** We introduce four different distance matrices to explore different K-Means variants and improve clustering performance on nonlinearly separable and complex data.

## 2. Related Work

### 2.1. K-Means

Given data $\mathbf{X} = \{\mathbf{x}_1, \mathbf{x}_2, \cdots, \mathbf{x}_n\} \in \mathbb{R}^{d \times n}$, the number of cluster centroids $c$, and the cluster centroids $\mathbf{u}_k = \dfrac{1}{n_k} \sum_{i=1}^{n_k} \mathbf{x}_i, (k = 1, 2, \ldots, c)$, $n_k$ is the number of samples in the $k$-th cluster, the K-Means clustering algorithm aims to minimize the sum of squared distances between each sample and its corresponding cluster centroid. The objective function of K-Means can be expressed as:

$$\min_{\mathbf{u}_k, \mathbf{F}} \sum_{k=1}^{c} \sum_{i=1}^{n_k} \|\mathbf{x}_i - \mathbf{u}_k\|_2^2 \mathbf{F}_{ik} \quad \text{s.t.} \quad \mathbf{F} \in \text{Ind} \quad (1)$$

where $\mathbf{F} \in \mathbb{R}^{n \times c}$ is a cluster cluster indicator matrix. The $i$-th row $\mathbf{f}^i$ of matrix $\mathbf{F}$ represents the label vector for the $i$-th sample. $\mathbf{F}_{ik} = 1$ indicates that the $i$-th sample belongs to the $k$-th cluster, and $\mathbf{F}_{ik} = 0$ otherwise. This means that each sample's label vector $\mathbf{f}^i$ is a one-hot label.

### 2.2. Kernel K-Means

Kernel K-Means is a generalization of the traditional K-Means clustering algorithm. First, a nonlinear mapping function $\phi$ is used to map the data points to a higher-dimensional feature space, and then the traditional K-Means algorithm is applied. Taking the Gaussian kernel K-Means as an example, its objective function can be expressed as:

$$\min_{\mathbf{u}_k, \mathbf{F}} \sum_{k=1}^{c} \sum_{i=1}^{n_k} \|\phi(\mathbf{x}_i) - \mathbf{u}_k\|_2^2 \mathbf{F}_{ik} \quad \text{s.t.} \quad \mathbf{F} \in \text{Ind} \quad (2)$$

where $\mathbf{u}_k = \dfrac{1}{n_k} \sum_{i=1}^{n_k} \phi(\mathbf{x}_i)$ is the centroid of the $k$-th cluster in the feature space. By employing the Gaussian kernel function $K(\mathbf{x}, \mathbf{y}) = \exp\left(-\dfrac{\|\mathbf{x} - \mathbf{y}\|^2}{2\sigma^2}\right)$, $\sigma$ is a scaling parameter, we can implicitly compute the inner product or distance in the high-dimensional feature space without the need to explicitly calculate the feature vector $\phi(\mathbf{x})$. The objective is to minimize the sum of squared distances between each data point and its assigned cluster centroid in the feature space, which allows for the discovery of nonlinear structures of the data.

### 2.3. Fuzzy K-Means

Fuzzy K-Means is an extension of the traditional K-Means algorithm and diverges from the hard clustering of the traditional K-Means by employing soft clustering, which permits each data point to belong to multiple clusters with a certain

degree of membership. This approach offers enhanced flexibility, particularly adept at handling scenarios where the boundaries between data points are not distinctly defined. The objective function of fuzzy K-Means can be articulated as follows:

$$\min_{\mathbf{u}_k, \mathbf{F}} \sum_{k=1}^{c} \sum_{i=1}^{n_k} \|\mathbf{x}_i - \mathbf{u}_k\|_2^2 \mathbf{F}_{ik}^m \quad \text{s.t. } \mathbf{F} \geqslant \mathbf{0}, \mathbf{F}\mathbf{1} = \mathbf{1} \quad (3)$$

where $\mathbf{1}$ is an all-one vector; $m > 1$ is a parameter that controls the fuzziness of the membership function; a larger $m$ indicates higher fuzziness.

## 3. A Unified Centerless Framework

The K-Means method and its variants described above rely on the calculation of cluster centers, making them susceptible to outliers, which can negatively affect clustering results. To address this issue, we first introduce Theorem 1, transforming the K-Means clustering into a label-guided manifold learning version, which is also equal to a unified centerless framework.

**Theorem 1.** *Let $\mathbf{x}_i$ be the $i$-th sample of data matrix $\mathbf{X}$, $\mathbf{u}_k$ be the centroid of the $k$-th cluster. Then, we have*

$$\sum_{k=1}^{c} \sum_{i=1}^{n_k} \|\mathbf{x}_i - \mathbf{u}_k\|_2^2 \mathbf{F}_{ik} = \sum_{i=1}^{n} \sum_{j=1}^{n} \|\mathbf{x}_i - \mathbf{x}_j\|_2^2 \mathbf{S}_{ij}, \quad (4)$$

*where $\mathbf{S}$ is the similarity matrix that represents the manifold structure of the data. It is constructed from the cluster indicator matrix $\mathbf{F}$, such that $\mathbf{S} = \mathbf{G}\mathbf{G}^\top$, where $\mathbf{G} = \mathbf{F}\mathbf{P}^{-1/2}$, $\mathbf{P} \in \mathbb{R}^{c \times c}$ is a diagonal matrix, and $\mathbf{P}_{kk} = \sum_{i=1}^{n} \mathbf{F}_{ik}$*

*Proof.* Expanding the left side of Equation (4) yields:

$$\begin{aligned}
&\text{tr}\left( \sum_{i=1}^{n} \sum_{k=1}^{c} \mathbf{x}_i \mathbf{x}_i^\top \mathbf{F}_{ik} \right) - 2\text{tr}\left( \sum_{i=1}^{n} \sum_{k=1}^{c} \mathbf{x}_i^\top \mathbf{u}_k \mathbf{F}_{ik} \right) \\
&+ \text{tr}\left( \sum_{i=1}^{n} \sum_{k=1}^{c} \mathbf{u}_k \mathbf{u}_k^\top \mathbf{F}_{ik} \right).
\end{aligned} \quad (5)$$

Taking the partial derivative with respect to $\mathbf{u}_k$ and setting it to zero, we find:

$$\mathbf{u}_k = \frac{\sum_{i=1}^{n} \mathbf{x}_i \mathbf{F}_{ik}}{\mathbf{P}_{kk}} = \mathbf{X}\mathbf{f}_k \mathbf{P}_{kk}^{-1}. \quad (6)$$

where $\mathbf{f}_k$ is the $k$-th column vector of $\mathbf{F}$.

Substituting Equation (6) back into Equation (5) and letting $\mathbf{Q} \in \mathbb{R}^{n \times n}$ be a diagonal matrix where $\mathbf{Q}_{ii} = \sum_{k=1}^{c} \mathbf{F}_{ik}$, we see that Equation (5) is simplified to:

$$\begin{aligned}
&\text{tr} \sum_{i=1}^{n} \mathbf{x}_i \mathbf{x}_i^\top \mathbf{Q}_{ii} - \text{tr} \sum_{k=1}^{c} \mathbf{X}\mathbf{f}_k \mathbf{P}_{kk}^{-1} \mathbf{f}_k^\top \mathbf{X}^\top \\
&= \text{tr}(\mathbf{X}(\mathbf{Q} - \mathbf{F}\mathbf{P}^{-1}\mathbf{F}^\top)\mathbf{X}^\top)
\end{aligned} \quad (7)$$

Letting the similarity matrix $\mathbf{S} = \mathbf{F}\mathbf{P}^{-1}\mathbf{F}^\top$, then

$$\mathbf{S}\mathbf{1} = \mathbf{F}\mathbf{P}^{-1}(\mathbf{1}^\top \mathbf{F})^\top = \mathbf{F}\mathbf{1} = \mathbf{Q}\mathbf{1} \quad (8)$$

Equation (8) means that $\mathbf{Q}$ is a degree matrix of $\mathbf{S}$, then Equation (7) can be written as:

$$\text{tr}(\mathbf{X}(\mathbf{Q} - \mathbf{F}\mathbf{P}^{-1}\mathbf{F}^\top)\mathbf{X}^\top) = \sum_{i=1}^{n} \sum_{j=1}^{n} \|\mathbf{x}_i - \mathbf{x}_j\|_2^2 \mathbf{S}_{ij} \quad (9)$$

Therefore, according to Equation (5), Equation (7), and Equation (9), we can conclude that Equation (4) holds and Theorem 1 is proved. $\square$

Since $\mathbf{S} = \mathbf{G}\mathbf{G}^\top$, $\mathbf{S}_{ij} = \langle \mathbf{g}^i, \mathbf{g}^j \rangle$, $\mathbf{g}^i$ is the $i$-th row of $\mathbf{G}$, if the elements of the distance matrix $\mathbf{D}$ are defined as $\mathbf{D}_{ij} = \|\mathbf{x}_i - \mathbf{x}_j\|_2^2$, then we have

$$\min_{\mathbf{F}} \sum_{i=1}^{n} \sum_{j=1}^{n} \|\mathbf{x}_i - \mathbf{x}_j\|_2^2 \mathbf{S}_{ij} = \min_{\mathbf{F}} \text{tr}(\mathbf{G}^\top \mathbf{D}\mathbf{G}) \quad (10)$$

According to Equation (4), we have:

$$\min_{\mathbf{u}_k, \mathbf{F}} \sum_{k=1}^{c} \sum_{i=1}^{n_k} \|\mathbf{x}_i - \mathbf{u}_k\|_2^2 \mathbf{F}_{ik} = \min_{\mathbf{F}} \text{tr}(\mathbf{G}^\top \mathbf{D}\mathbf{G}) \quad (11)$$

Drawing from the Equation (11), we have derived a unified centerless framework for K-Means and manifold learning. This framework not only dispenses with the calculation of cluster centroids but also employs a cluster indicator matrix to construct a similarity matrix $\mathbf{S}$, thereby preserving the consistency between the data's manifold structure and its labels. In what follows, we will discuss the **new version of K-Means, Kernel K-Means, and Fuzzy K-Means**.

**(1) K-Means**: According to Theorem 1, the objective of traditional K-Means can be reformulated as follows:

$$\begin{aligned}
\min_{\mathbf{F}} \sum_{i=1}^{n} \sum_{j=1}^{n} &\|\mathbf{x}_i - \mathbf{x}_j\|_2^2 \mathbf{S}_{ij} = \min_{\mathbf{F}} \text{tr}(\mathbf{G}^\top \mathbf{D}\mathbf{G}) \\
&= \min_{\mathbf{F}} \text{tr}((\mathbf{F}\mathbf{P}^{-1/2})^\top \mathbf{D}(\mathbf{F}\mathbf{P}^{-1/2})) \\
&= \min_{\mathbf{F}} \text{tr}(\mathbf{F}^\top \mathbf{D}\mathbf{F}(\mathbf{P})^{-1}) \quad \text{s.t. } \mathbf{F} \in \text{Ind}
\end{aligned} \quad (12)$$

where $\mathbf{F} \in \mathbb{R}^{n \times c}$ denotes the cluster indicator matrix, and $\mathbf{D} \in \mathbb{R}^{n \times n}$ represents the distance matrix.

**(2) Kernel K-Means:** When applied Theorem 1 in the kernel space, Equation (2) can be rewritten as

$$\begin{aligned}
\min_{\mathbf{F}} \sum_{i=1}^{n} \sum_{j=1}^{n} &\|\phi(\mathbf{x}_i) - \phi(\mathbf{x}_j)\|_2^2 \mathbf{S}_{ij} = \min_{\mathbf{F}} \text{tr}(\mathbf{G}^\top \mathbf{D}\mathbf{G}) \\
&= \min_{\mathbf{F}} \text{tr}(\mathbf{F}^\top \mathbf{D}\mathbf{F}(\mathbf{F}^\top \mathbf{F})^{-1}) \quad \text{s.t. } \mathbf{F} \in \text{Ind}
\end{aligned} \quad (13)$$

where $\mathbf{D}_{ij} = \|\phi(\mathbf{x}_i) - \phi(\mathbf{x}_j)\|_2^2 = K(\mathbf{x}_i, \mathbf{x}_i) + K(\mathbf{x}_j, \mathbf{x}_j) - 2K(\mathbf{x}_i, \mathbf{x}_j)$. The novel form Equation (13) of Kernel K-Means avoids reliance on cluster centroids, enhancing the method's capacity to handle nonlinear data and imparting greater robustness of clustering results.

**(3) Fuzzy K-Means:** Let $\mathbf{W}_{ik} = \mathbf{F}_{ik}^m$, $\mathbf{G} = \mathbf{W}\mathbf{P}^{-1/2}$, $\mathbf{S} = \mathbf{G}\mathbf{G}^\top$, and $\mathbf{P} \in \mathbb{R}^{c \times c}$ be a diagonal matrix where $\mathbf{P}_{kk} = \sum_{i=1}^n \mathbf{W}_{ik}$. According to Theorem 1, Equation (3) can be reformulated as

$$\min_{\mathbf{F}} \sum_{i=1}^n \sum_{j=1}^n \|\mathbf{x}_i - \mathbf{x}_j\|_2^2 \mathbf{S}_{ij} = \min_{\mathbf{F}} \text{tr}(\mathbf{G}^\top \mathbf{D}\mathbf{G})$$
$$= \min_{\mathbf{F}} \text{tr}(\mathbf{W}^\top \mathbf{D}\mathbf{W}(\mathbf{P})^{-1}) \quad \text{s.t. } \mathbf{F} \geqslant \mathbf{0}, \mathbf{F}\mathbf{1} = \mathbf{1} \quad (14)$$

where the elements of the distance matrix $\mathbf{D}$ are defined as $\mathbf{D}_{ij} = \|\mathbf{x}_i - \mathbf{x}_j\|_2^2$.

Therefore, we can say that K-Means, Kernel K-Means, and Fuzzy K-Means are all unified in our centerless manifold framework $\min_{\mathbf{F}} \text{tr}(\mathbf{G}^\top \mathbf{D}\mathbf{G})$.

# 4. Methodology

## 4.1. Motivation and Objective

We substitute $\mathbf{G} = \mathbf{F}\mathbf{P}^{-1/2}$ into Equation (10) and rewrite the unified K-Means as $\min_{\mathbf{F}} \text{tr}(\mathbf{F}^\top \mathbf{D}\mathbf{F}(\mathbf{P})^{-1})$. Since the cluster indicator matrix $\mathbf{F}$ is discrete and challenging to optimize directly, and it does not ensure a balanced clustering, we introduce a balanced regularization term based on $\ell_{2,1}$-norm and obtain the overall objective:

$$\min_{\mathbf{F}} \text{tr}(\mathbf{F}^\top \mathbf{D}\mathbf{F}(\mathbf{P})^{-1}) - \lambda \|\mathbf{F}^\top\|_{2,1} \quad \text{s.t. } \mathbf{F} \geqslant \mathbf{0}, \mathbf{F}\mathbf{1} = \mathbf{1} \quad (15)$$

where $\lambda$ is a balance parameter. Then, we prove that maximizing $\|\mathbf{F}^\top\|_{2,1}$ helps to address both issues via Theorem 2.

**Theorem 2.** *Given $n_1 + n_2 + \ldots + n_c = n$, where $n_k \geq 0$ represents the number of samples in the $k$-th cluster. By achieving the following balanced regularization term, $\mathbf{F}$ is discrete and exhibits a balanced cluster distribution, i.e., the maximum will be attained when $n_1 = n_2 = \ldots = n_c$.*

$$\max_{\mathbf{F}} \|\mathbf{F}^\top\|_{2,1} \quad \text{s.t. } \mathbf{F} \geqslant \mathbf{0}, \mathbf{F}\mathbf{1} = \mathbf{1} \quad (16)$$

*Proof.* Let $\mathbf{f}_k \in \mathbb{R}^{n \times 1}$ represents $k$-th column of $\mathbf{F}$. The $\ell_{2,1}$-norm of the cluster indicator matrix $\mathbf{F}^\top \in \mathbb{R}^{c \times n}$ is defined as follows:

$$\|\mathbf{F}^\top\|_{2,1} = \sum_{k=1}^c \sqrt{\mathbf{f}_k^\top \mathbf{f}_k} = \sum_{k=1}^c \sqrt{\sum_{i=1}^n \mathbf{F}_{ik}^2} = \sum_{k=1}^c a_k \quad (17)$$

where

$$a_k = \sqrt{\sum_{i=1}^n \mathbf{F}_{ik}^2} = \sqrt{\mathbf{f}_k^\top \mathbf{f}_k} \quad (18)$$

**Lemma 1.** *According to the Cauchy-Schwarz inequality, let $\mathbf{a} = [a_1, a_2, \ldots, a_c]^\top \in \mathbb{R}^{c \times 1}$, $\mathbf{e} = [1, 1, \cdots, 1]^\top \in \mathbb{R}^{c \times 1}$. Then, we have*

$$|\langle \mathbf{a}, \mathbf{e} \rangle| \leq \|\mathbf{a}\|_2 \|\mathbf{e}\|_2 \Rightarrow \sum_{k=1}^c a_k \leq \|\mathbf{a}\|_2 \sqrt{(1 + 1 + \cdots + 1)}$$
$$\Rightarrow \sum_{k=1}^c a_k \leq \|\mathbf{a}\|_2 \sqrt{c} \quad (19)$$

*The equality holds if and only if $a_1 = a_2 = \cdots = a_c$.*

According to Lemma 1, maximizing $\|\mathbf{F}^\top\|_{2,1} = \sum_{k=1}^c a_k$ is equivalent to maximizing $\|\mathbf{a}\|_2$. By substituting into Equation (18), we obtain

$$\|\mathbf{a}\|_2 = \sqrt{\sum_{k=1}^c a_k^2} = \sqrt{\sum_{k=1}^c \sum_{i=1}^n \mathbf{F}_{ik}^2} = \sqrt{\sum_{i=1}^n \sum_{k=1}^c \mathbf{F}_{ik}^2} \quad (20)$$

Thus, we have

$$\max \|\mathbf{a}\|_2 \Rightarrow \max \|\mathbf{a}\|_2^2 = \max \sum_{i=1}^n \sum_{k=1}^c \mathbf{F}_{ik}^2 \quad (21)$$

Obviously, each row $\mathbf{f}^i$ of $\mathbf{F}$ is independent, so for each row, we have

$$\max_{\mathbf{F}_{ik}} \sum_{k=1}^c \mathbf{F}_{ik}^2 \quad \text{s.t. } 0 \leq \mathbf{F}_{ik} \leq 1, \sum_{k=1}^c \mathbf{F}_{ik} = 1 \quad (22)$$

The solution to maximizing Equation (22) should be realized when $\mathbf{f}^i$ has only one element equal to 1 and the rest are 0, and the maximum value of Equation (22) should be 1. Thus, we can conclude that Equation (22) of $n$ rows reach maximum only when $\mathbf{F}$ is a discrete cluster indicator matrix.

In this case, $\mathbf{F}^\top \mathbf{F} \in \mathbb{R}^{c \times c}$ is a diagonal matrix whose $k$-th diagonal element is the number of samples in the $k$-th cluster, hence:

$$a_k = \sqrt{\mathbf{f}_k^\top \mathbf{f}_k} = \sqrt{n_k} \quad (23)$$

where $n_k$ is the number of samples of the $k$-th cluster, $\mathbf{f}_k$ is the $k$-th column of the matrix $\mathbf{F}$.

According to $a_1 = a_2 = \cdots = a_c$, we have $\sqrt{n_1} = \sqrt{n_2} = \cdots = \sqrt{n_c}$, and the equality holds if and only if $n_1 = n_2 = \cdots = n_c = n/c$. $\square$

Theorem 2 demonstrates that Equation (16) can achieve an approximate cluster balance. When the optimal solution is achieved, $(\mathbf{F}^\top \mathbf{F})^{1/2} = \sqrt{\frac{n}{c}}\mathbf{I}$. Consequently, our objective function in the Equation (15) can be simplified to:

$$\min_{\mathbf{F}} \text{tr}(\mathbf{F}^\top \mathbf{D}\mathbf{F}) - \lambda \|\mathbf{F}^\top\|_{2,1} \quad \text{s.t. } \mathbf{F} \geqslant \mathbf{0}, \mathbf{F}\mathbf{1} = \mathbf{1} \quad (24)$$

When Equation (24) achieves the optimal solution, $\mathbf{F}$ is discrete and each cluster is balanced.

## 4.2. Optimization

The $\ell_{2,1}$-norm in Equation (24) is difficult to solve directly by gradient descent because of its non-smooth property. To simplify the optimization process, we define $f(\mathbf{F}) = \|\mathbf{F}^\top\|_{2,1}$ and perform a first-order Taylor expansion at $\mathbf{F}^{(t)}$ as follows:

$$f(\mathbf{F}) = f(\mathbf{F}^{(t)}) + \langle \nabla f(\mathbf{F}^{(t)}), \mathbf{F} - \mathbf{F}^{(t)} \rangle \qquad (25)$$

where $\mathbf{F}^{(t)}$ is the solution at the $t$-th iteration, and $\nabla f(\mathbf{F}^{(t)})$ is the gradient of $\|\mathbf{F}^\top\|_{2,1}$. The derivative of $\|\mathbf{F}^\top\|_{2,1}$ with respect to $\mathbf{F}$ is denoted as $\mathbf{H}$, given by:

$$\mathbf{H} = \frac{\partial \|\mathbf{F}^\top\|_{2,1}}{\partial \mathbf{F}} = \frac{\frac{1}{2}\mathrm{tr}\left(\mathbf{F}\boldsymbol{\Lambda}\mathbf{F}^\top\right)}{\partial \mathbf{F}} = \mathbf{F}\boldsymbol{\Lambda} \qquad (26)$$

where $\boldsymbol{\Lambda}$ is a diagonal matrix and $\boldsymbol{\Lambda}_{kk} = \frac{1}{\|\mathbf{f}_k\|_2}$; $\mathbf{f}_k \in \mathbb{R}^{n \times 1}$ is the $k$-th column of $\mathbf{F}$.

Ignoring the constant in the Equation (25), we solve the Equation (24) iteratively as follows

$$\begin{aligned}\mathbf{F}^{(t+1)} &= \min_{\mathbf{F}} \ \mathrm{tr}(\mathbf{F}^\top\mathbf{D}\mathbf{F}) - \lambda\langle \nabla f(\mathbf{F}^{(t)}), \mathbf{F}\rangle \\ &= \min_{\mathbf{F}} \ \mathrm{tr}(\mathbf{F}^\top\mathbf{D}\mathbf{F}) - \lambda\mathrm{tr}(\mathbf{H}^\top\mathbf{F})\end{aligned} \qquad (27)$$

So we approximate Equation (24) to Equation (28), $\mathbf{F}$ is updated by solving the following problem:

$$\min_{\mathbf{F}\geqslant 0, \mathbf{F1}=1} \mathrm{tr}(\mathbf{F}^\top\mathbf{D}\mathbf{F}) - \lambda\mathrm{tr}(\mathbf{H}^\top\mathbf{F}) \qquad (28)$$

Let $\tilde{\mathbf{F}} = \begin{bmatrix}\mathbf{f}^i \\ \mathbf{F}_0\end{bmatrix}$, $\tilde{\mathbf{D}} = \begin{bmatrix}\mathbf{D}_{ii} & \mathbf{d}_{i0}^\top \\ \mathbf{d}_{i0} & \mathbf{D}_{00}\end{bmatrix}$, where $\mathbf{f}^i \in \mathbb{R}^{1\times c}$ is the $i$-th row of the matrix $\mathbf{F}$, $\mathbf{F}_0 \in \mathbb{R}^{(n-1)\times c}$ represents the matrix formed by all rows of matrix $\mathbf{F}$ except for the $i$-th row $\mathbf{f}^i$. $\mathbf{d}_{i0} \in \mathbb{R}^{(n-1)\times 1}$ denotes the column vector consisting of all elements of the $i$-th column $\mathbf{d}_i$ excluding the element $\mathbf{D}_{ii}$ of matrix $\mathbf{D}$. Lastly, $\mathbf{D}_{00} \in \mathbb{R}^{(n-1)\times(n-1)}$ represents the matrix formed by all elements of matrix $\mathbf{D}$ except for those in the $i$-th row and $i$-th column. Obviously, $\mathrm{tr}(\mathbf{F}^\top\mathbf{D}\mathbf{F}) = \mathrm{tr}(\tilde{\mathbf{F}}^\top\tilde{\mathbf{D}}\tilde{\mathbf{F}})$, and we have:

$$\begin{aligned}\tilde{\mathbf{F}}^\top\tilde{\mathbf{D}}\tilde{\mathbf{F}} &= \begin{bmatrix}(\mathbf{f}^i)^\top & (\mathbf{F}_0)^\top\end{bmatrix}\begin{bmatrix}\mathbf{D}_{ii} & \mathbf{d}_{i0}^\top \\ \mathbf{d}_{i0} & \mathbf{D}_{00}\end{bmatrix}\begin{bmatrix}\mathbf{f}^i \\ \mathbf{F}_0\end{bmatrix} \\ &= (\mathbf{f}^i)^\top\mathbf{D}_{ii}\mathbf{f}^i + (\mathbf{F}_0)^\top\mathbf{d}_{i0}\mathbf{f}^i + (\mathbf{f}^i)^\top\mathbf{d}_{i0}^\top\mathbf{F}_0 \\ &\quad + (\mathbf{F}_0)^\top\mathbf{D}_{00}\mathbf{F}_0\end{aligned} \qquad (29)$$

Let $\tilde{\mathbf{H}} = \begin{bmatrix}\mathbf{h}^i \\ \mathbf{H}_0\end{bmatrix}$, $\mathbf{h}^i$ is the $i$-th row of $\mathbf{H}$, $\mathbf{H}_0 \in \mathbb{R}^{(n-1)\times c}$ represents the matrix formed by all rows of matrix $\mathbf{H}$ except

for the $i$-th row $\mathbf{h}^i$, $\tilde{\mathbf{H}}^\top\tilde{\mathbf{F}} = \begin{bmatrix}(\mathbf{h}^i)^\top & (\mathbf{H}_0)^\top\end{bmatrix}\begin{bmatrix}\mathbf{f}^i \\ \mathbf{F}_0\end{bmatrix} = (\mathbf{h}^i)^\top\mathbf{f}^i + (\mathbf{H}_0)^\top\mathbf{F}_0$. Note that $\mathrm{tr}(\mathbf{H}^\top\mathbf{F}) = \mathrm{tr}(\tilde{\mathbf{H}}^\top\tilde{\mathbf{F}})$, and we have:

$$\begin{aligned}\tilde{\mathbf{F}}^\top\tilde{\mathbf{D}}\tilde{\mathbf{F}} - \lambda\tilde{\mathbf{H}}^\top\tilde{\mathbf{F}} =& (\mathbf{f}^i)^\top\mathbf{D}_{ii}\mathbf{f}^i + (\mathbf{F}_0)^\top\mathbf{d}_{i0}\mathbf{f}^i \\ &+ (\mathbf{f}^i)^\top\mathbf{d}_{i0}^\top\mathbf{F}_0 + (\mathbf{F}_0)^\top\mathbf{D}_{00}\mathbf{F}_0 \\ &- \lambda(\mathbf{h}^i)^\top\mathbf{f}^i - \lambda(\mathbf{H}_0)^\top\mathbf{F}_0\end{aligned} \qquad (30)$$

Then, removing items not related to variable $\mathbf{f}^i$, through the properties of trace operation, we have:

$$\begin{aligned}&\mathrm{tr}(\mathbf{F}^\top\mathbf{D}\mathbf{F} - \lambda\mathbf{H}^\top\mathbf{F}) \\ =&\mathrm{tr}((\mathbf{f}^i)^\top\mathbf{D}_{ii}\mathbf{f}^i + 2\mathbf{f}^i\mathbf{F}_0^\top\mathbf{d}_{i0} - \lambda\mathbf{f}^i(\mathbf{h}^i)^\top) \\ =&\mathbf{f}^i(\mathbf{f}^i)^\top\mathbf{D}_{ii} + \mathbf{f}^i(2\mathbf{F}_0^\top\mathbf{d}_{i0} - \lambda(\mathbf{h}^i)^\top)\end{aligned} \qquad (31)$$

Thus, the problem of updating the $i$-th row of $\mathbf{F}$ can be:

$$\min_{\mathbf{f}^i\mathbf{1}=1} \mathbf{f}^i(\mathbf{f}^i)^\top\mathbf{D}_{ii} + \mathbf{f}^i(2\mathbf{F}_0^\top\mathbf{d}_{i0} - \lambda(\mathbf{h}^i)^\top) \qquad (32)$$

As $\mathbf{D}_{ii} = 0 (i = 1, 2, \cdots, n)$, Equation (32) can be:

$$\min_{\mathbf{f}^i} \mathbf{f}^i(2\mathbf{F}_0^\top\mathbf{d}_{i0} - \lambda(\mathbf{h}^i)^\top) \Leftrightarrow \min_{\mathbf{f}^i} \mathbf{f}^i(2\mathbf{F}^\top\mathbf{d}_i - \lambda(\mathbf{h}^i)^\top) \qquad (33)$$

where $\mathbf{d}_i$ is the $i$-th column of $\mathbf{D}$, $\mathbf{D}_{ii} = 0$. $\mathbf{F}$ denotes the solution before $\mathbf{f}^i$ is updated. Then, the solution of $\mathbf{f}^i$ can be:

$$\mathbf{F}_{ib} = \begin{cases}1, & b = \arg\min_{k}(2\mathbf{F}^\top\mathbf{d}_i - \lambda(\mathbf{h}^i)^\top)_k \\ 0, & \text{otherwise.}\end{cases} \qquad (34)$$

The detailed optimization for $\mathbf{F}$ is summarized in Appendix A.

## 4.3. Complexity Analysis

When updating the $i$-th row of $\mathbf{F}$, we compute $\mathbf{F}^\top\mathbf{d}_i - \lambda(\mathbf{h}^i)^\top$, which requires a time complexity of $\mathcal{O}(nc)$ and $c$ addition operations. Thus, updating all rows of $\mathbf{F}$ necessitates a complexity of $\mathcal{O}(n^2c)$, with $nc$ addition operations. Therefore, the algorithm requires $\mathcal{O}(n^2c + nc)$ for each iteration. To address this, we propose an acceleration strategy to reduce the time complexity to $\mathcal{O}(nc)$ during the iteration process. Our algorithm's acceleration strategy and pseudo-code flow can be found in the Appendix A.

## 5. Four methods of calculating distance matrix

In order to demonstrate the applicability of our method to a variety of different datasets, we present the following four methods of calculating distance matrices:

**1. Euclidean square distance:** $\mathbf{D}_{ij} = \|\mathbf{x}_i - \mathbf{x}_j\|_2^2$.

**2. $K$ nearest neighbor distance:**

$$\mathbf{D}_{ij} = \begin{cases} \|\mathbf{x}_i - \mathbf{x}_j\|_2^2, & \mathbf{x}_i \in N_k(\mathbf{x}_j) \text{ or } \mathbf{x}_j \in N_k(\mathbf{x}_i) \\ \sigma, & \text{otherwise} \end{cases}$$

where $\sigma$ is a large constant, $N_k(\mathbf{x}_i)$ is the $K$ nearest neighbors of the sample $\mathbf{x}_i$.

**3. Gaussian kernel distance:** $\mathbf{D}_{ij} = \|\phi(\mathbf{x}_i) - \phi(\mathbf{x}_j)\|_2^2 = K(\mathbf{x}_i, \mathbf{x}_i) + K(\mathbf{x}_j, \mathbf{x}_j) - 2K(\mathbf{x}_i, \mathbf{x}_j)$, where $\phi(\mathbf{x})$ represents the nonlinear mapping of sample $\mathbf{x}$, and the Gaussian Kernel is $K(\mathbf{x}, \mathbf{y}) = \exp\left(-\dfrac{\|\mathbf{x} - \mathbf{y}\|^2}{2\sigma^2}\right)$, $\sigma$ is a scaling parameter.

**4. Low-pass filtering distance:** In addition to the three distances mentioned above, we have also introduced a low-pass filter distance. A low-pass filter allows low frequencies to pass and blocks high frequencies. With a cutoff frequency $\omega_c = \dfrac{1}{2\pi}$, its amplitude-frequency characteristic is given by $|H(j\omega)| = \dfrac{1}{\sqrt{1 + (\omega \cdot 2\pi)^2}}$. Inspired by this characteristic, when the similarity matrix $\mathbf{R}$ is used as input, if we want to retain samples with high similarity and filter out samples with low similarity to address the issue of nonlinear separability, we can rewrite the distance matrix as $\mathbf{D}_{ij} = a|H(j\omega)|^2$, $\omega = \mathbf{R}_{ij}$, where $a$ is a hyperparameter. Through this filtering function, we can see that the higher the similarity, the smaller the distance, and the lower the similarity, the bigger the distance. This retains samples with high similarity, filters out samples with low similarity, and effectively preserves the intrinsic manifold structure of the data, which can better handle nonlinear separable data.

## 6. Experiments

In this chapter, we conducted relevant experiments on two toy datasets and ten benchmark datasets, and selected 6 classic clustering comparison algorithms for comparison. Our experiments were conducted on a Windows 11 system, 13th Gen Intel(R) Core(TM) CPU, and MATLAB R2023a.

### 6.1. Benchmark datasets

**Datasets:** We selected the following ten datasets: CMUPIE (Sim et al., 2002), digits (Kusetogullari et al., 2020), FERET (Phillips et al., 2000), Mpeg7 (Bober, 2001), olivetti (Samaria & Harter, 1994), Palm[1], Pengdigits (Liu & Wechsler, 1997), PEAL (Wang & Tang, 2004), STL10 (Coates et al., 2011), and USPS (Hull, 1994). Detailed information on the datasets is provided in the Appendix A.

**Comparison Methods:** We selected the following six comparison methods: K-Means (Hartigan & Wong, 1979), KKM

---

[1]https://www.scholat.com/xjchensz

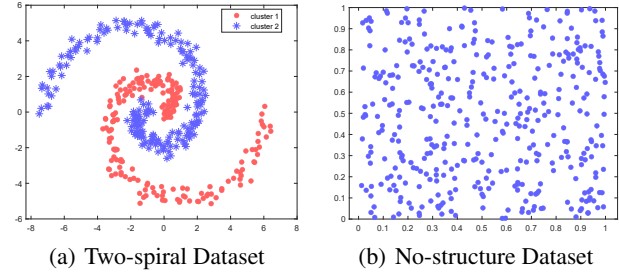

(a) Two-spiral Dataset      (b) No-structure Dataset

*Figure 1.* Toy datasets.

(Tzortzis & Likas, 2008), RKM (Lin et al., 2019), CDKM (Nie et al., 2022), K-sum (Pei et al., 2023), and K-sum-x (Pei et al., 2023).

### 6.2. Clustering performance

The experimental results of our method utilizing four distance matrices and six comparative algorithms on ten benchmark datasets are presented in Tables 1 to 3. The parameter values of our model are in the Appendix A. It is evident that algorithms such as K-Means, RKM, and CDKM, which rely on the initialization of cluster centroids, are susceptible to the influence of these centroids. Additionally, their use of Euclidean squared distance impedes their ability to effectively handle complex data structures, resulting in inferior clustering performance compared to K-sum and K-sum-X, which leverage neighborhood relationships and do not require cluster centroid initialization. However, these latter approaches operate under the assumption of balanced datasets, potentially limiting their applicability to certain specific data distributions.

In contrast, our four distance-based models employ a centerless approach, all of which have yielded notably improved results. Particularly, our low-pass filtering distance metric, which harnesses a nonlinear mapping to transform the data's similarity relationships into distance relationships, has demonstrated exceptional performance. This low-pass filtering distance preserves the manifold structure of the data and the consistency of the labels, where higher similarity corresponds to shorter distances and lower similarity corresponds to larger distances. Consequently, when confronted with non-linearly separable datasets, our low-pass filtering distance-based model exhibits superior clustering outcomes.

### 6.3. Toy datasets

To verify the feasibility of our method on nonlinear datasets and demonstrate that our method can achieve cluster balance, we only created the following two datasets.

**Two-spiral Dataset:** There are a total of four hundred samples, divided into two clusters, with 200 samples in each

*Table 1.* The clustering performances on the CMUPIE, digits, FERET, and Mpeg7 datasets.

| Datasets | CMUPIE | | | digits | | | FERET | | | Mpeg7 | | |
|---|---|---|---|---|---|---|---|---|---|---|---|---|
| Methods | ACC | NMI | Purity | ACC | NMI | Purity | ACC | NMI | Purity | ACC | NMI | Purity |
| K-Means | 0.1968 | 0.4121 | 0.2145 | 0.4353 | 0.4439 | 0.5966 | 0.2651 | 0.6514 | 0.2975 | 0.4557 | 0.6595 | 0.4856 |
| KKM | 0.1915 | 0.3770 | 0.2118 | 0.4368 | 0.4526 | 0.6008 | 0.2157 | 0.5807 | 0.2407 | 0.4993 | 0.6892 | 0.5236 |
| RKM | 0.1740 | 0.4182 | 0.1859 | 0.4042 | 0.4181 | 0.5833 | 0.3121 | 0.7090 | 0.3236 | 0.5264 | 0.7126 | 0.5521 |
| CDKM | 0.1994 | 0.4146 | 0.2187 | 0.4338 | 0.4438 | 0.5975 | 0.2834 | 0.6819 | 0.3168 | 0.5091 | 0.7196 | 0.5440 |
| K-sum | 0.2150 | 0.4764 | 0.2283 | 0.1390 | 0.3416 | **0.7422** | 0.2879 | 0.6993 | 0.2921 | 0.5386 | 0.7257 | 0.5721 |
| K-sum-x | 0.1754 | 0.4208 | 0.1824 | 0.1412 | 0.3417 | 0.7405 | 0.2936 | 0.7056 | 0.3079 | 0.5393 | 0.7288 | 0.5629 |
| Our-ED | 0.1765 | 0.4200 | 0.1891 | 0.3855 | 0.4195 | 0.5395 | 0.3000 | 0.7071 | 0.3086 | **0.5714** | **0.7463** | **0.5979** |
| Our-KNN | 0.3634 | 0.6375 | 0.3736 | 0.4570 | 0.4636 | 0.6058 | 0.3193 | 0.7183 | 0.3321 | 0.5600 | 0.7280 | 0.5800 |
| Our-K-ED | 0.2171 | 0.4844 | 0.2297 | 0.4550 | 0.4461 | 0.5848 | 0.3107 | 0.7117 | 0.3207 | 0.5650 | 0.7413 | 0.5900 |
| Our-LF | **0.6341** | **0.8174** | **0.6436** | **0.5073** | **0.5076** | 0.6305 | **0.3257** | **0.7233** | **0.3343** | 0.4600 | 0.6602 | 0.4821 |

*Table 2.* The clustering performances on the olivetti, Palm, USPS, and Pendigits datasets.

| Datasets | olivetti | | | Palm | | | USPS | | | Pendigits | | |
|---|---|---|---|---|---|---|---|---|---|---|---|---|
| Methods | ACC | NMI | Purity | ACC | NMI | Purity | ACC | NMI | Purity | ACC | NMI | Purity |
| K-Means | 0.4488 | 0.4697 | 0.4856 | 0.7047 | 0.8947 | 0.7558 | 0.6458 | 0.6026 | 0.7129 | 0.6963 | 0.6705 | 0.7260 |
| KKM | 0.4900 | 0.4862 | 0.5233 | 0.6620 | 0.8732 | 0.7065 | 0.6872 | 0.6437 | 0.7565 | 0.7859 | 0.7139 | 0.7859 |
| RKM | 0.4922 | 0.4671 | 0.5144 | 0.7690 | 0.9180 | 0.7825 | 0.6241 | 0.5748 | 0.7003 | 0.7296 | 0.6639 | 0.7296 |
| CDKM | 0.4617 | 0.4795 | 0.4896 | 0.7225 | 0.9052 | 0.7745 | 0.6526 | 0.6094 | 0.7237 | 0.7027 | 0.6697 | 0.7226 |
| K-sum | 0.4233 | 0.4282 | 0.4656 | 0.8405 | 0.9405 | 0.8520 | 0.6802 | 0.6274 | 0.7486 | 0.7562 | 0.6743 | 0.7562 |
| K-sum-x | 0.4322 | 0.4099 | 0.4467 | 0.8145 | 0.9321 | 0.8295 | 0.6502 | 0.5853 | 0.7150 | 0.7768 | 0.7001 | 0.7768 |
| Our-ED | 0.5078 | 0.4910 | 0.5256 | 0.8460 | 0.9401 | 0.8555 | 0.6539 | 0.5842 | 0.7164 | 0.7816 | 0.7056 | 0.7816 |
| Our-KNN | 0.5767 | 0.5361 | 0.5933 | 0.8585 | 0.9456 | 0.8665 | 0.7545 | 0.6690 | 0.7545 | 0.8406 | 0.7719 | 0.8406 |
| Our-K-ED | 0.5478 | 0.5229 | 0.5700 | 0.8325 | 0.9362 | 0.8440 | 0.7595 | 0.6559 | 0.7595 | **0.8579** | **0.7784** | **0.8579** |
| Our-LF | **0.8322** | **0.8181** | **0.8322** | **0.9055** | **0.9674** | **0.9105** | **0.8450** | **0.7803** | **0.8450** | 0.8322 | 0.7612 | 0.8322 |

*Table 3.* The clustering performances on the PEAL and STL10 datasets.

| Datasets | PEAL | | | STL10 | | |
|---|---|---|---|---|---|---|
| Methods | ACC | NMI | Purity | ACC | NMI | Purity |
| K-Means | 0.7206 | 0.8939 | 0.7539 | 0.8088 | 0.8049 | 0.8328 |
| KKM | 0.7087 | 0.8624 | 0.7296 | 0.9162 | 0.8548 | 0.9162 |
| RKM | 0.8072 | 0.9129 | 0.8181 | 0.9422 | 0.8796 | 0.9422 |
| CDKM | 0.7296 | 0.8967 | 0.7617 | 0.8094 | 0.8053 | 0.8333 |
| K-sum | 0.8770 | 0.9424 | 0.8811 | 0.9220 | 0.8502 | 0.9220 |
| K-sum-x | 0.8491 | 0.9291 | 0.8537 | 0.9215 | 0.8505 | 0.9215 |
| Our-ED | 0.8596 | 0.9321 | 0.8640 | 0.9212 | 0.8500 | 0.9212 |
| Our-KNN | **0.8919** | **0.9417** | **0.8939** | 0.9198 | 0.8357 | 0.9198 |
| Our-K-ED | 0.8602 | 0.9317 | 0.8649 | 0.9228 | 0.8516 | 0.9228 |
| Our-LF | 0.8854 | 0.9446 | 0.8889 | **0.9539** | **0.8969** | **0.9539** |

cluster. As can be seen from Figure 1.(a), this is a nonlinearly separable data, which can be used to verify the ability of our method to handle nonlinearly separable data. As shown in Figure 2, the results of K-Means, KKM, CDKM, and our method on the double spiral dataset. K-Means and

CDKM use Euclidean distance and are affected by the initialization of cluster centroids, so they cannot effectively separate the nonlinearly separable dataset. Although KKM uses the kernel method to calculate, it still relies on the calculation of cluster centroids, while our method does not require the calculation of cluster centroids and can well maintain the inherent structure of the data.

**No-structure Dataset:** A dataset of 400 samples without structure as shown in Figure 1.(b), with each sample randomly distributed on this two-dimensional plane, which can be used to verify that our method can ensure cluster balance during clustering. For the unstructured dataset, we conducted binary clustering, three-cluster clustering, four-cluster clustering, and five-cluster clustering, as shown in Figure 3. Regardless of how many clusters the unstructured dataset is divided into, cluster balance can be achieved, with the number of samples in each cluster basically remaining consistent.

Under the ideal cluster balance condition, when $n = 400$ samples are divided into $c = 2, 3, 4, 5$ clusters, the sample number of each cluster should be $n/c = 200, 133, 100, 80$.

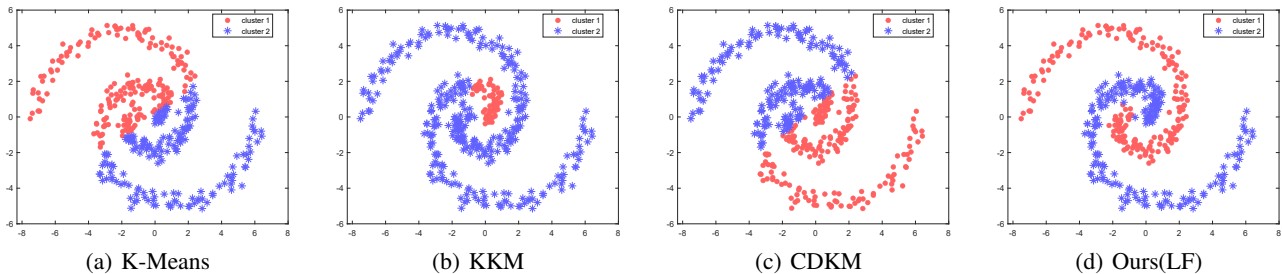

| (a) K-Means | (b) KKM | (c) CDKM | (d) Ours(LF) |

*Figure 2.* The clustering results of our method and three comparison methods K-Means, KKM and CDKM on the two-spiral dataset.

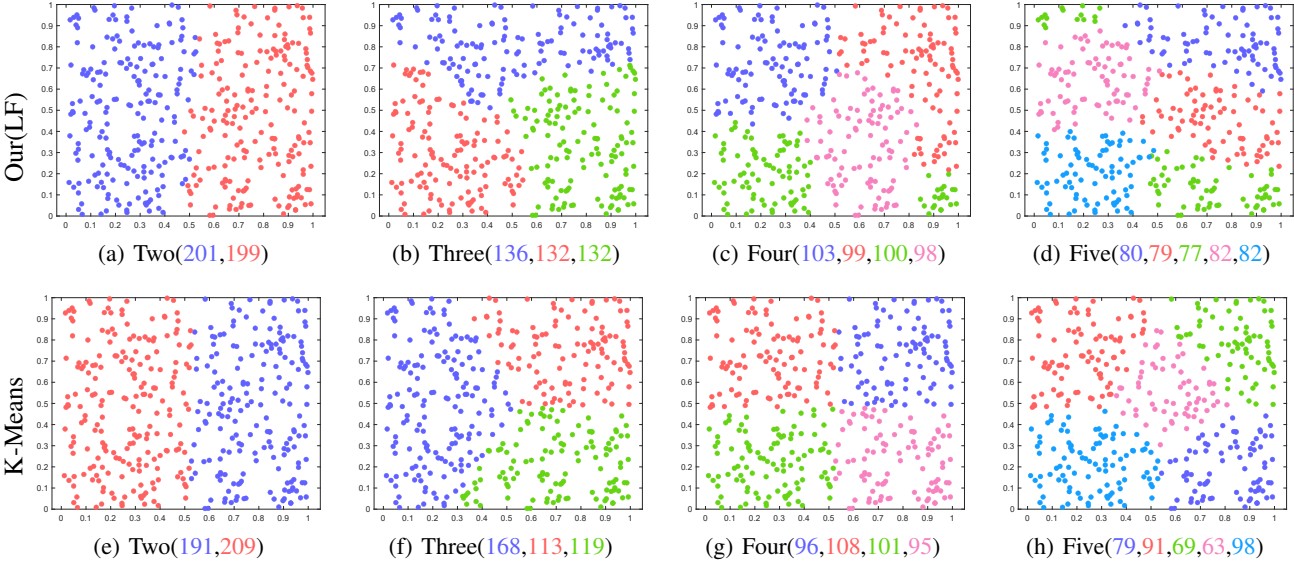

| (a) Two(201,199) | (b) Three(136,132,132) | (c) Four(103,99,100,98) | (d) Five(80,79,77,82,82) |

| (e) Two(191,209) | (f) Three(168,113,119) | (g) Four(96,108,101,95) | (h) Five(79,91,69,63,98) |

*Figure 3.* The clustering results of our method on the no-structure dataset, different colors represent different clusters.

*Table 4.* The value of SSe.

| Methods | Two clusters | Three clusters | Four clusters | Five clusters |
|---------|--------------|----------------|---------------|---------------|
| Ours (LF) | 2 | 11 | 14 | 18 |
| K-Means | 162 | 1821 | 106 | 856 |

If we make a quantitative analysis, and calculate every clustering, the error sum of squares of each cluster is $SSe = \sum_{k=1}^{c} \left(n_k - \frac{n}{c}\right)^2$, the larger the value, the worse the balance, the smaller the value, the better the balance. The specific calculation results can be seen in Table 4.

Moreover, from Figure 3.(d) and (e), we can see that our clustering is not a simple linear clustering, and a cluster is not just around a cluster centroid, which also indicates that our method has good anti-noise performance.

**More experiments, including T-SNE visualization, parameter and convergence analysis, can be found in the**

**Appendix A.**

# 7. Conclusion

In this paper, we introduce a novel unified K-Means clustering framework that achieves balanced clustering and eliminates the need for computing cluster centroids. Specifically, it first unifies the multiple versions of K-Means into a label-guided manifold learning version, which utilizes the cluster indicator matrix to construct the similarity matrix and exploit the discriminative manifold structure guided by pseudo labels. Furthermore, it incorporates $\ell_{2,1}$-norm based balanced regularization to enforce producing a more robust and balanced clustering outcome. Four different distance metrics are introduced to enhance the method's flexibility and adaptivity to different data distributions. Especially, the adopted low-pass filtering distance effectively handles nonlinear data structures. Finally, experiments on both toy and benchmark datasets demonstrate that our method outperforms comparison approaches.

## Acknowledgements

This work is supported by the National Natural Science Foundation of China under Grant 62176203, the Fundamental Research Funds for the Central Universities (ZYTS25267, QTZX25004), and the Science and Technology Project of Xi'an (Grant 2022JH-JSYF-0009), Open Project of Anhui Provincial Key Laboratory of Multimodal Cognitive Computation, Anhui University (No. MMC202416), Selected Support Project for Scientific and Technological Activities of Returned Overseas Chinese Scholars in Shaanxi Province 2023-02, and the Xidian Innovation Fund (Project NoYJSJ25007).

## Impact Statement

This paper presents work whose goal is to advance the field of Machine Learning. There are many potential societal consequences of our work, none which we feel must be specifically highlighted here.

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

# A. Appendix

## A.1. Acceleration strategy

To enhance the algorithm's efficiency, we calculate and retain the values of $2\mathbf{F}^\top \mathbf{d}_i - \lambda(\mathbf{h}^i)^\top$ for $i = 1, 2, \ldots, n$, which requires $\mathcal{O}(n^2)$ time complexity. This allows the resolution of each row, as referenced in Equation (34), to be completed with only $c$ addition operations, resulting in a cumulative total of $nc$ additions for each iteration.

Before updating the $i$-th row $\mathbf{f}^i$ of $\mathbf{F}$, we record the index $p$ corresponding to the non-zero element in $\mathbf{f}^i$. Then, we calculate Equation (34) to obtain $b$. If $b = p$, no changes are necessary; however, if $b$ differs from $p$, the adjustments to $2\mathbf{f}_k^\top \mathbf{D} - \lambda \mathbf{h}_k^\top$ for $k = p, b$ are expressed as follows:

$$2\mathbf{f}_p^\top \mathbf{D} - \lambda \mathbf{h}_p^\top \leftarrow 2\mathbf{f}_p^\top \mathbf{D} - \lambda \mathbf{h}_p^\top - 2\mathbf{d}_i, \quad 2\mathbf{f}_b^\top \mathbf{D} - \lambda \mathbf{h}_b^\top \leftarrow 2\mathbf{f}_b^\top \mathbf{D} - \lambda \mathbf{h}_b^\top + 2\mathbf{d}_i \tag{35}$$

where $\mathbf{f}_p$ is the $p$-th column of $\mathbf{F}$, $\mathbf{f}_b$ is the $b$-th column of $\mathbf{F}$. $\mathbf{h}_p$ is the $p$-th column of $\mathbf{H}$, $\mathbf{h}_b$ is the $b$-th column of $\mathbf{H}$. $\mathbf{d}_i$ is the $i$-th column of $\mathbf{D}$. This process involves just two additional operations. Each iteration for this step takes $2v$ additions, where $v \leq n$. Consequently, this method reduces the time complexity per iteration from $\mathcal{O}(n^2 c + nc)$ to $\mathcal{O}(nc + 2v)$, which simplifies to $\mathcal{O}(nc)$.

Overall, this approach significantly minimizes the computational load of the algorithm, improving its efficiency. The comprehensive flow of the algorithm can be found in Algorithms 1 and 2.

---

**Algorithm 1** Optimizing $\mathbf{F}$

---

**Input:** distance matrix $\mathbf{D} \in \mathbb{R}^{n \times n}$, matrix $\mathbf{H}$, hyperparameter $\lambda$.
**Initialize:** cluster indicator matrix $\mathbf{F} \in \mathbb{R}^{n \times c}$
Calculate and store $2\mathbf{F}^\top \mathbf{D} - \lambda \mathbf{H}^\top$, $n_k = \mathbf{f}_k^\top \mathbf{1}$
**while** $\mathbf{F}$ not converge **do**
  **for** $i = 1$ $to$ $n$ **do**
    Update $p$ by the index of element 1 in $\mathbf{f}^i$
    **if** $n_p = 1$ **then**
      continue
    **end if**
    Compute $b = \underset{k}{\mathrm{argmin}}(2\mathbf{F}^\top \mathbf{d}_i - \lambda(\mathbf{h}^i)^\top)_k$
    **if** $b \neq p$ **then**
      Update $2\mathbf{f}_k^\top \mathbf{D} - \lambda \mathbf{h}_k^\top (k = p, b)$ via Equation (35)
    **end if**
  **end for**
**end while**
**Output** $\mathbf{F} \in \mathbb{R}^{n \times c}$

---

---

**Algorithm 2** Solving problem (24)

---

**Input:** distance matrix $\mathbf{D} \in \mathbb{R}^{n \times n}$, cluster number $c$, hyperparameter $\lambda$.
**Initialize:** cluster indicator matrix $\mathbf{F} \in \mathbb{R}^{n \times c}$
**while** not converge **do**
  Update matrix $\mathbf{H}$ by Equation (26);
  Update matrix $\mathbf{F}$ by Algorithm 1;
**end while**
**Output** $\mathbf{F} \in \mathbb{R}^{n \times c}$

---

## A.2. Convergence Analysis

To address the non-smooth issue of the $\ell_{2,1}$-norm in the objective function (24), we performed a Taylor expansion on $f(\mathbf{F}) = \|\mathbf{F}^\top\|_{2,1}$ and then solved for the cluster indicator matrix $\mathbf{F}$ through multiple iterations. The $\mathbf{F}^{(t+1)}$ obtained at the $(t+1)$-th iteration can be derived from:

$$\mathbf{F}^{(t+1)} = \min_{\mathbf{F}} \mathrm{tr}(\mathbf{F}^\top \mathbf{D} \mathbf{F}) - \lambda \mathrm{tr}(\mathbf{H}^\top \mathbf{F}) \tag{36}$$

Since $\mathbf{D}$ is a positive definite matrix and $\mathrm{tr}(\mathbf{H}^\top \mathbf{F})$ is a linear function of $\mathbf{F}$, the formula is convex in $\mathbf{F}$. Because $\mathbf{F}$ is a discrete cluster indicator matrix with elements valued at 0 or 1, and $\mathbf{D}$ and $\mathbf{H}$ are fixed matrices, the value of the formula varies within a finite range. Furthermore, by updating $\mathbf{F}$ through solving the minimization problem, the value of the objective function does not increase. Specifically, after each update, the value of the objective function either remains unchanged or decreases. All iterative results are located in the compact set $\{\mathbf{F} \mid \mathbf{F1} = \mathbf{1}, \mathbf{F} \geqslant \mathbf{0}\}$. Therefore, our model is convergent and will obtain an optimal solution during the iteration process.

### A.3. The information of benchmark datasets

*Table 5.* The information of datasets.

| Datasets | Samples | Features | Clusters | Type |
| --- | --- | --- | --- | --- |
| CMUPIE | 2856 | 1024 | 68 | images |
| digits | 4000 | 256 | 10 | images |
| FERET | 1400 | 6400 | 200 | images |
| Mpeg7 | 1400 | 6000 | 70 | images |
| olivetti | 900 | 2500 | 10 | images |
| Palm | 2000 | 256 | 100 | images |
| Pendigits | 10,992 | 16 | 10 | images |
| PEAL | 30,863 | 256 | 1,040 | images |
| STL10 | 13000 | 2048 | 10 | images |
| USPS | 9,298 | 256 | 10 | images |

### A.4. Parameter setting

Since Equation (24) contains the parameter $\lambda$, we need to set the parameter $\lambda$ for all four distances. In addition, Our-KNN needs to set the value of the nearest neighbor $K$; using the Gaussian kernel function $K(\mathbf{x}, \mathbf{y}) = \exp\left(-\dfrac{\|\mathbf{x} - \mathbf{y}\|^2}{2\sigma^2}\right)$, the Our-K-ED contains the parameter $\sigma$.

While utilizing the Low-pass filtering distance defined as $\mathbf{D}_{ij} = a\left|H(j\omega)\right|^2, \quad \omega = \mathbf{R}_{ij}$, we employ the directly alternate sampling (DAS) method (Li et al., 2022) to select $r$ anchor points (where $r < n$), denoted as $\mathbf{A} \in \mathbb{R}^{d \times r}$. The number of selecting anchor points is configured according to the ratio range provided in (Li et al., 2022), where the anchor ratio is from 0.1 to 0.9 with a step of 0.1. Subsequently, we apply the method from (Nie et al., 2016) to obtain the corresponding anchor graph $\mathbf{T} \in \mathbb{R}^{n \times r}$. Let $\rho$ represent the number of neighboring anchor points for each sample; the specific computation formula for the anchor graph $\mathbf{T}$ is as follows:

$$\mathbf{T}_{ij} = \begin{cases} \frac{h(\mathbf{x}_i, \mathbf{a}_{\rho+1}) - h(\mathbf{x}_i, \mathbf{a}_j)}{\rho h(\mathbf{x}_i, \mathbf{a}_{r+1}) - \sum_{z=1}^{\rho} h(\mathbf{x}_i, \mathbf{a}_z)} & j \leq \rho \\ 0 & j > \rho \end{cases} \tag{37}$$

where $h(\mathbf{x}_i, \mathbf{y}_j) = \|\mathbf{x}_i - \mathbf{y}_j\|_2^2$, and $\mathbf{a}_j$ represents the $j$-th neighboring anchor point. Finally, based (Liu et al., 2010) and the anchor graph $\mathbf{T}$, we can derive the input matrix $\mathbf{R}$ for the Low-pass filtering distance as:

$$\mathbf{R} = \mathbf{T}\Delta^{-1}\mathbf{T}^\top \tag{38}$$

where $\Delta \in \mathbb{R}^{r \times r}$ has diagonal elements defined as $\Delta_{jj} = \sum_{i=1}^{n} \mathbf{T}_{ij}$, with all other elements being zero.

- **Our-ED:** The values of $\lambda$ for CMUPIE, digits, FERET, Mpeg7, olivetti, Palm, Pengdigits, PEAL, STL10, and USPS are 7696000, 2886000, 1000, 1200, 1801000, 13126000, 500000, 88000, 500000, 52600.

- **Our-KNN:** The values of $(K, \lambda)$ for CMUPIE, digits, FERET, Mpeg7, olivetti, Palm, Pengdigits, PEAL, STL10, and USPS are (20, 15000000), (100, 300000), (12, 500000), (19, 2000), (80, 300000), (10, 100000), (712, 100000), (29, 80000), (1000, 80000), (445, 50000).

- **Our-K-ED:** The values of $(\sigma, \lambda)$ for CMUPIE, digits, FERET, Mpeg7, olivetti, Palm, Pengdigits, PEAL, STL10, and USPS are (0.1, 0.001), (0.2, 9), (0.2, 0.001), (1, 0.8), (0.5, 9), (0.8, 0.1), (0.09, 500), (16, 0.04), (0.5, 0.001), (0.1, 0.7).

- **Our-LF:** The values of $\lambda$ for CMUPIE, digits, FERET, Mpeg7, olivetti, Palm, Pengdigits, PEAL, STL10, and USPS are 0.43, 0.71, 0.57, 0.23, 0.2, 0.74, 0.6, 0.15, 0.1, 0.82.

### A.5. T-SNE visualization on the USPS dataset

We have visualized the performance of various methods on the USPS dataset, as shown in Figure 4.(a) depicts the true cluster labels, while Figure 4.(b)-(g) present the results of the six comparative algorithms. The final subfigure showcases our method based on the low-pass filtering distance metric.

Examining the true label representation in Figure 4.(a), we observe that clusters 4 and 6, as well as clusters 5 and 10, are relatively proximal to one another. Among the six comparative methods, regardless of whether they are based on Euclidean distance or KNN distances, they struggle to accurately recognize the underlying cluster relationships between these two pairs of closely related clusters. In contrast, our low-pass filtering distance-based clustering approach is able to effectively differentiate these proximal clusters, and the resulting clustering outcomes closely align with the true label distribution, exhibiting the most favorable performance.

This visual analysis highlights the ability of our low-pass filtering distance metric to better preserve the inherent manifold structure and cluster proximities present in the USPS dataset, leading to superior clustering outcomes compared to the alternative algorithms considered.

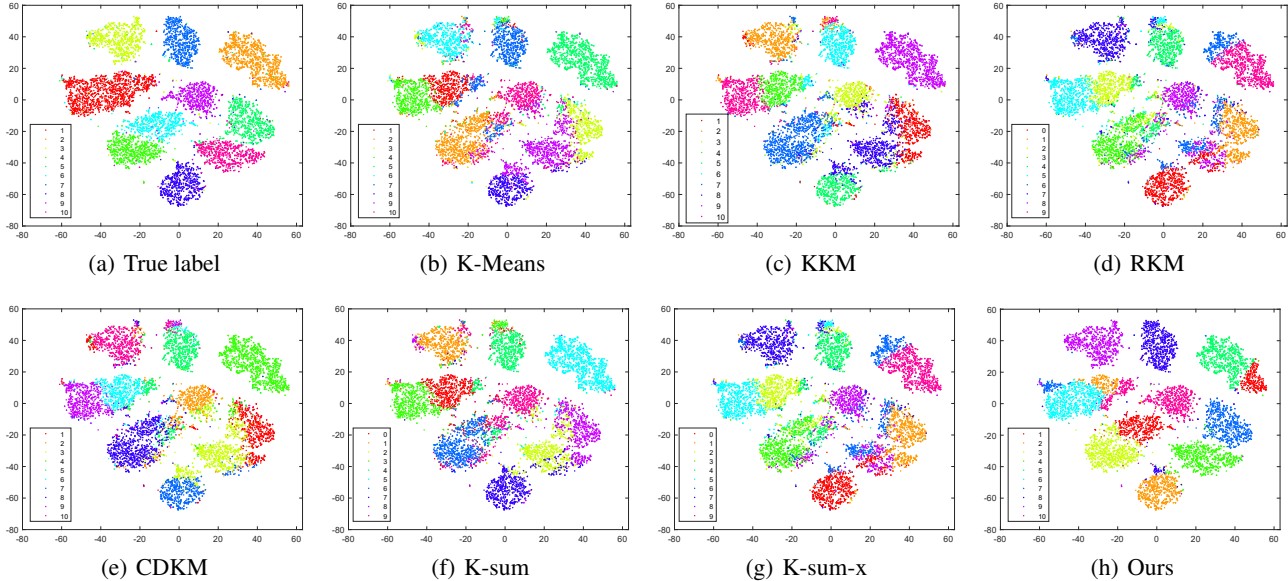

*Figure 4.* T-SNE Visualization of the clustering effect of our method and six comparison methods on the USPS database.

### A.6. Parameter analysis

In Figure 5, we further discuss the impact of the parameter $\lambda$ associated with the balanced regularization term on the clustering performance across different datasets. We selected four datasets - CMUPIE, digits, olivetti and Palm - to investigate this relationship.

The results indicate that the clustering performance exhibits varying trends as a function of $\lambda$ for the different datasets. However, a emerges where the clustering performance first increases and then decreases as $\lambda$ is increased. This suggests that the selection of an appropriate value for $\lambda$ is crucial, as overly large or small values can lead to suboptimal clustering outcomes.

The observed trends highlight the importance of dataset-specific parameter tuning to achieve the best clustering performance. The careful selection of the $\lambda$ parameter, which controls the sparsity and regularization of the model, plays a significant role in the effectiveness of the clustering algorithms across diverse data domains.

## A.7. Convergence

As shown in Figure 6, our method exhibits rapid convergence on the four evaluated datasets. The final objective function values reach a balanced and stable state, and the clustering performance also gradually stabilizes with increasing iteration counts. This observation demonstrates the strong convergence properties of our algorithm and the effectiveness of the optimization process underlying our algorithm.

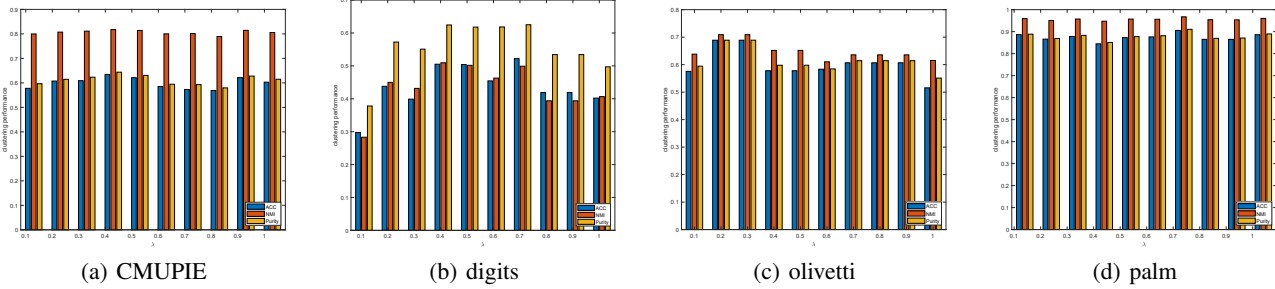

| (a) CMUPIE | (b) digits | (c) olivetti | (d) palm |

*Figure 5.* The evaluations of clustering performance with parameter $\lambda$ on CMUPIE and digits datasets.

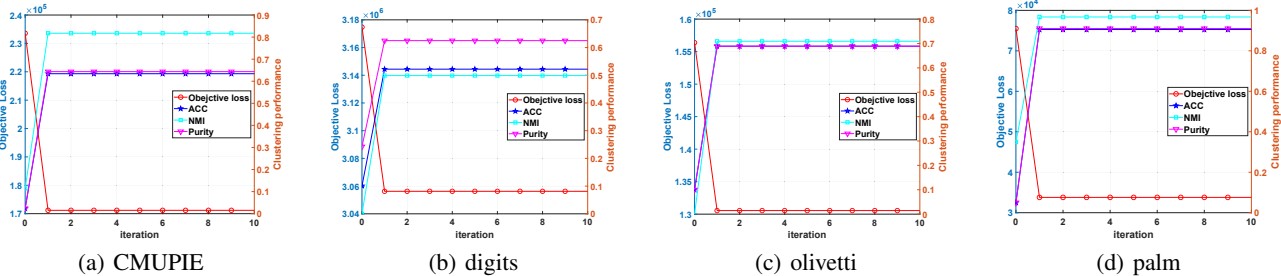

| (a) CMUPIE | (b) digits | (c) olivetti | (d) palm |

*Figure 6.* Objective loss and clustering performance with iterations on CMUPIE and digits datasets.

