# OpenReview forum: "Unified K-Means Clustering with Label-Guided Manifold Learning"
_ICML.cc/2025/Conference — ICML 2025 poster_

### Official Review · Reviewer_YTpU · 2025-03-07

**Overall Recommendation:** 4

**Summary:**

This paper introduces a framework that integrates K-means clustering, Kernel K-means clustering, and Fuzzy K-means clustering with manifold learning. This framework aims to tackle the challenges of initial centroid sensitivity, handling nonlinear datasets, and achieving balanced clustering in traditional K-means clustering.

**Claims And Evidence:**

Yes.

**Essential References Not Discussed:**

In both the introduction and the comparison algorithms, the authors discussed essential relevant works.

**Experimental Designs Or Analyses:**

Yes. The experimental design of the work follows existing works, and the analyses are overall reasonable.

**Methods And Evaluation Criteria:**

Yes. The experiments on benchmark datasets and toy datasets demonstrate the method's excellent clustering performance, its ability to handle nonlinear data, and its capacity for balanced clustering.

**Other Comments Or Suggestions:**

There are some typos, such as the comparison algorithm K-sum-x being written as K-sum-X in Section 6.2.

**Other Strengths And Weaknesses:**

This paper introduces a unified framework that addresses both the sensitivity of K-means to cluster centers and its inability to handle nonlinearity, and proposes a low-pass filtering distance metric. It achieves superior experimental results. The shortcomings can be found in the comments and questions.

**Questions For Authors:**

1. Formulas (20) to (24) are not very logical; please restate them.
2. Is the label matrix F a discrete hard label or a continuous label?
3. The issue associated with K-means that is sensitive to centroid initialization is a well-recognized challenge, and has been studied by several works. What differentiates the proposed method from existing approaches, and what advantages does it offer?

**Relation To Broader Scientific Literature:**

The unified framework proposed by the authors addresses both the sensitivity of K-means to cluster centers and its inability to handle nonlinearity, as well as the issue of balanced clustering.

**Theoretical Claims:**

Yes, the paper presents several theoretical claims, and the theoretical claims in this paper are overall correct and supported by the corresponding proof.

---

> ### Author Rebuttal · Authors · 2025-03-31
>
> Thank you very much for your recognition and valuable comments. We provide the following responses according to your questions:
>
> **1. Explain the formula (20) to (24)**
>
> **A**: We compute the $\ell_{2,1}$-norm of the cluster indicator matrix $\mathbf{F}^\top$ and then maximize this term to achieve balanced clustering. Equations (20) to (24) are used to prove that when maximizing the $\ell_{2,1}$-norm of matrix $\mathbf{F}^\top$, its optimal solution is a discrete matrix $\mathbf{F} $. We will make a detail explanation for them.
>
> **2. Is label matrix F a discrete hard label or a continuous label**
>
> **A**: We have demonstrated that problem (24) reaches a maximum when $\mathbf{F}$ is a discrete indicator matrix, and thus, the final optimized matrix $\mathbf{F}$ is a discrete hard label.
>
> **3. The difference of the proposed method from existing works addressing centroid initialization sensitivity and its advantages**
>
> **A**: To avoid the impact of center initialization on K-means clustering, methods such as K-means++, K-sum, and K-sum-X have been proposed. K-means++ randomly selects a point from the dataset as the first cluster center. For all unselected points in the dataset, it calculates the minimum Euclidean distance to the currently selected center and then randomly selects the next cluster center based on the squared distance as a probability weight, until convergence. Although K-means++ alleviates the sensitivity of K-means to initial centers to some extent, it still relies on the computation of cluster centers and may be affected by outliers. K-sum and K-sun-X, on the other hand, do not directly compute cluster centers but perform clustering based on the relationships between data points. While these two methods address the sensitivity of K-means to initial centers, they do not consider balanced clustering. Compared with these methods, our methods introduces the $\ell_{2,1}$-norm regularization term to prevent certain clusters from containing too few or too many data samples, addressing the impact of imbalanced datasets on the model and achieving more balanced clustering results.

---

> > ### Comment · Reviewer_YTpU · 2025-04-04
> >
> > The authors' responses have addressed my concerns, and I have checked the responses to the questions of all the reviewers and have no further questions. Therefore, I decide to raise the rating accordingly.

---

> > > ### Author Response · Authors · 2025-04-06
> > >
> > > Thank you for your support and recognition of our work. We will carefully revise the final version based on your valuable feedback to further improve the quality of the paper. We sincerely appreciate the thoughtful suggestions you have provided and the time and effort you have dedicated to this process.

---

### Official Review · Reviewer_DEHu · 2025-03-08

**Overall Recommendation:** 4

**Summary:**

In this paper, a novel K-Means clustering was proposed, named unified K-Means clustering with label-guided manifold learning, to solve the problems of traditional K-Means algorithm, such as the sensitivity of initial centroid selection, the limited recognition ability of intrinsic manifold structure of nonlinear data sets, and the difficulty of achieving balanced clustering in practical scenarios. The method does not need to calculate cluster centers and uses labels to guide the exploration of the nonlinear structure of the data. It realizes the cluster balance with L2,1-norm and achieves high clustering accuracy and robustness. Extensive experiments are performed, whose results prove the superiority of the proposed method.

**Claims And Evidence:**

Yes

**Essential References Not Discussed:**

The related works covered in this paper are relatively comprehensive, involving traditional k-means, fuzzy k-means, kernel k-means for handling nonlinear data, balanced clustering method RKM, and k-means with centered variants CDKM and k-sum and k-sum-x without centers, etc.

**Experimental Designs Or Analyses:**

Yes, the authors selected 10 multi-view datasets and 6 comparison methods, using ACC, NMI and Purity as clustering measures. Data sets range from small data sets to large data sets with sample sizes of 10,000 and 30,000. The comparison method also includes the SOTA variants of K-Means. There is no obvious problem with the experimental designs or analyses.

**Methods And Evaluation Criteria:**

Yes

**Other Comments Or Suggestions:**

1. What is the significance of balanced clustering? The authors should emphasize this issue in Introduction.
2. The authors should carefully check the paper. For example, in Equation (16), it seems matrix F is missing in some minimization operations.
3. The unified framework is defined as min tr(GᵀDG), but the final model is min tr(FᵀDF) without the matrix P. The authors should provide a detailed explanation for this issue.

**Other Strengths And Weaknesses:**

Strengths:
1. The authors proposed a novel K-Means clustering, named unified K-Means clustering with label-guided manifold learning, which addresses the weakness of K-means that is sensitive to the initialization of cluster centers and achieves a broader application on nonlinear data.
2. The authors imposed a L2,1-norm constraint on the clustering indicator matrix, allowing the model to achieve class balance during clustering. Detailed logical reasoning and proof processes were provided to enhance readability.
3. The authors also employed different distance matrices to enhance the model's generalization ability.
Weaknesses:
1. The paper is on the basis of manifold learning but provides little discussion on how it is connected to manifold learning and its advantages.

**Questions For Authors:**

1.In Equation (14), matrix G is constructed from matrices F and P. Why is matrix P not present in the final expression? What is the relationship between matrix P and matrix F?
2. How is the parameter a for the low-pass filtering distance chosen?

**Relation To Broader Scientific Literature:**

The paper's main contribution is the unified K-Means clustering with label-guided manifold learning and diverse distance metrics. It enhances robustness and achieves broader applications on nonlinear data, supported by comprehensive experiments that validate the method's effectiveness.

**Theoretical Claims:**

Yes, I reviewed the correctness of the proofs for the theoretical claims presented in the paper, specifically focusing on Theorem 3.1 and Theorem 4.1, and there is no problem.

---

> ### Author Rebuttal · Authors · 2025-03-31
>
> Thank you very much for your recognition and valuable comments. Here are our responses:
>
> **1. Connection to manifold learning and its advantages**
>
> **A**: We have shown the equivalence between K-means and manifold learning by transforming K-means into the form of manifold learning using the cluster indicator matrix to construct the similarity matrix $\mathbf{S}$. This transformation eliminates the need to compute cluster centers and allows K-means to explore the local manifold structure of the data, making it suitable for nonlinear datasets.
>
> **2. The significance of balanced clustering**
>
> **A**: Many clustering algorithms like K-Means are based on distance tend to assign more data samples to larger clusters, which may lead to a decrease in algorithm performance because some datasets may contain outliers or noise. K-means may group the majority of samples into one cluster while assigning outliers to another cluster. Balanced clustering can avoid it by ensuring the sample number of each cluster to be roughly the same, thereby achieving better stability and reliability. In summary, balanced clustering enhances resistance to outliers.
>
> **3. The reason for that the final model is min tr(FᵀDF) without the matrix P, and the relationship between matrix P and matrix F**
>
> **A**: According to Theorem 3.1, $\mathbf{G} = \mathbf{F}\mathbf{P}^{-1/2}$, where $\mathbf{P} = diag(p_1, \ldots, p\_c)$ and $p\_k = \sum\_{i=1}^{n} \mathbf{F}\_{ik}$. Since $\mathbf{F}$ is a cluster indicator matrix, $\mathbf{F}\_{ik} = 1$ when the $i$-th sample belongs to the $k$-th cluster. Each column of the matrix represents a class. Therefore, we can say that the $k$-th element of matrix $\mathbf{P}$ is actually the number of samples in the $k$-th cluster, which can also be rewritten as $\mathbf{P} = \mathbf{F}^\top \mathbf{F}$. Our model implements balanced clustering, ensuring that the number of samples in each cluster is equal. Thus, $\mathbf{P} = \mathbf{F}^\top \mathbf{F} = \frac{n}{c}\mathbf{I}$, where $n$, $c$, and $\mathbf{I}$ represent the number of samples, the number of clusters, and the identity matrix, respectively. Ignoring the constant term, the optimization problem $\min tr(\mathbf{G}^\top \mathbf{D} \mathbf{G})$ becomes $\min tr(\mathbf{F}^\top \mathbf{D} \mathbf{F})$.
>
> **4. How to choose the parameter $a$ for the low-pass filtering distance**
>
> **A**: Our low-pass filtering distance assigns a smaller distance to samples with high similarity and a larger distance to samples with low similarity, effectively pulling similar samples closer and pushing dissimilar samples farther apart. The advantage is that it increases the discriminability of samples by making dissimilar samples more distant. The parameter $a$ controls the degree of this pulling and pushing. However, a higher value of $a$ is not always better because it may also push similar samples farther apart, which is not conducive to clustering. Therefore, we choose $a = 1$ or $2$ for our low-pass filtering distance to better handle nonlinearly separable data.

---

### Official Review · Reviewer_sTfJ · 2025-03-08

**Overall Recommendation:** 4

**Summary:**

The manuscript presents a new balanced k-means clustering framework based on manifold learning for k-means clustering problem. The framework formulates balanced k-means as an optimization problem about the clustering label matrix and realizes clustering in one step by minimizing the objective function. In this paper, it is proved that k-means is equivalent to manifold learning under certain circumstances. Unlike traditional k-means, the model does not require centroid initialization, which the authors claim could lead to more robust results. The convergence can be accelerated by a pre-computational strategy.

**Claims And Evidence:**

Yes.

**Essential References Not Discussed:**

No.

**Experimental Designs Or Analyses:**

Yes. The dataset used in the experiment covers a wide range of categories and quantities, enabling a comprehensive presentation of the results. Additionally, the comparison of experimental methods includes advanced models from the past three years, enhancing the credibility of the experimental results.

**Methods And Evaluation Criteria:**

Yes.

**Other Comments Or Suggestions:**

The subheadings of Figures 6(b), (c), and (d) contain errors and should be replaced with the correct dataset names.

**Other Strengths And Weaknesses:**

Strength: The advantage of this research lies in the innovative combination of two traditional machine learning methods, K-means and manifold learning, and the rigorous derivation process that demonstrates the scientific validity of the approach. Furthermore, its effectiveness is well demonstrated through comprehensive datasets and experimental comparisons.

Weakness: However, the explanation of the necessity of balanced clustering is somewhat lacking. Providing additional clarification on this aspect would enhance the readability of the paper.

**Questions For Authors:**

1.	In the comparative experiments, K-sum and K-sum-x should also be centerless clustering methods. Why does your method achieve better results?
2.	The \(\ell_{2,1}\) norm regularization is generally a non-convex problem. What techniques or methods are used in the paper to achieve the optimization objective?
3.	From the evaluation results on different datasets, why does the same method exhibit significant differences in performance across different datasets? What are the underlying reasons for this?

**Relation To Broader Scientific Literature:**

Compared to previous studies, the highlights of this research lie in centerless clustering, which effectively reduces the impact of outliers on clustering results. Additionally, by incorporating manifold learning, the method better handles nonlinear structured data. Lastly, the introduction of a balanced regularization term ensures more balanced clustering results and prevents trivial solutions.

**Theoretical Claims:**

Yes. The paper presents Theorems 3.1 and 4.1, both of which are accompanied by proofs. Theorem 3.1 establishes the equivalence between K-means and manifold learning, with a proof that is clear and well-structured, and Theorem 4.1 discusses the ability of this method that supports the balanced clustering.

---

> ### Author Rebuttal · Authors · 2025-03-31
>
> Thank you very much for your recognition and valuable comments. Here are our responses:
>
> **1. Necessity of balanced clustering**
>
> **A**: Many clustering algorithms like K-Means are based on distance tend to assign more data samples to larger clusters, which may lead to a decrease in algorithm performance because some datasets may contain outliers or noise. K-means may group the majority of samples into one cluster while assigning outliers to another cluster. Balanced clustering can avoid it by ensuring the sample number of each cluster to be roughly the same, thereby achieving better stability and reliability. In summary, balanced clustering enhances resistance to outliers.
>
> **2. The reason for better performance against other centerless methods (K-sum and K-sum-x)**
>
> **A**: K-sum is a clustering algorithm based on a k-NN graph, and K-sum-X is a variant of K-sum that does not rely on a k-NN graph but directly uses features for clustering. These two comparison algorithms do not require the initialization and computation of cluster centers. Compared with these two methods, our method further introduces a balance regularization term, which has been demonstrated to be effective to improve clustering performance. Thus, our method achieves better performance than these two methods.
>
> **3. The technique to optimize the non-convex $\ell_{2,1}$-norm regularization term**
>
> **A**: The $\ell_{2,1}$-norm regularization is indeed a non-smooth problem, making our final model (26) difficult to solve directly using coordinate descent. Therefore, we perform a first-order Taylor expansion on the regularization term and transmit it into an iterative problem as (29), which is a convex function with respect to matrix $\mathbf{F}$, successfully addressing the non-convex issue of $\ell_{2,1}$-norm regularization term.
>
> **4. The reason of same method’s different performance across different datasets**
>
> **A**: Datasets may have vastly different distributions and structures. Some datasets may have clear and separable clusters, while others may have overlapped or nested clusters. The complex structure of datasets can affect clustering performance. For example, the FERET and Mpeg7 datasets have dimensions of 6400 and 6000, respectively, and numbers of clusters of 200 and 70. Unsupervised clustering methods might not realize satisfactory performance when directly applied to these high-dimensional datasets with many clusters. For the balanced CMUPIE dataset, comparison algorithms achieve an average clustering accuracy of 0.1920, while our method, with the balance regularization term, reaches 0.3478. Overall, the different structures of datasets directly affect clustering performance, but our method, with the introduction of the balance term, performs better than other K-means clustering methods.

---

### Official Review · Reviewer_w5A2 · 2025-03-10

**Overall Recommendation:** 4

**Summary:**

This work introduces an innovative centerless K-means clustering framework combined with manifold learning to improve clustering robustness and accuracy. By eliminating centroid initialization and utilizing a label matrix for similarity computation, the proposed method aligns manifold structures with class labels. Additionally, four distance metrics, including low-pass filtering distance, are incorporated to enhance adaptability to complex data. Theoretical derivations and comprehensive experiments on benchmark datasets demonstrate the effectiveness of the approach.

**Claims And Evidence:**

Yes

**Essential References Not Discussed:**

I personally believe that the related work it mentions is quite comprehensive.

**Experimental Designs Or Analyses:**

Yes

**Methods And Evaluation Criteria:**

Yes

**Other Comments Or Suggestions:**

The authors should pay more attention to some formatting details. For example, equation (12) does not explain what the matrix D is, the notation for the trace (tr) in equation (16) is inconsistent, as it uses both text and character formats, along with a series of similar issues.

**Other Strengths And Weaknesses:**

This paper proposes a centerless K-Means clustering method based on manifold learning, which enhances clustering stability and robustness while improving adaptability to nonlinear data structures. Experimental results indicate that it outperforms traditional K-Means and some improved methods.

However, this approach relies on the selection of the parameter $\lambda$. Additionally, although its centerless strategy eliminates sensitivity to initial centroids, it may introduce extra computational overhead under certain data distributions.

**Questions For Authors:**

1.	How is the matrix F initialized, and what are the convergence conditions for Algorithm 1 and Algorithm 2?
2.	Why is there the conclusion that "The solution to the maximization problem (24) should be realized when F_i has only one element equal to 1 and the rest are 0, and the maximum value should be 1. Thus, we can conclude that the problem (24) only reaches a maximum when F is a discrete label matrix"?
3.	The input to the low-pass filter distance is a graph; does the graph construction process introduce additional computational complexity?

**Relation To Broader Scientific Literature:**

K-means clustering has gained widespread attention for its simplicity and effectiveness. However, it inevitably struggles with nonlinear data and requires cluster centroid initialization, making it susceptible to noise. This paper reconstructs the K-means objective based on label-guided manifold learning, effectively addressing these shortcomings in the relevant field.

**Theoretical Claims:**

Yes

---

> ### Author Rebuttal · Authors · 2025-03-31
>
> Thank you very much for your recognition and valuable comments. Here are our responses:
>
> **1. The method relies on selection of $\lambda$**
>
> **A**: In the model proposed in this paper, the parameter $\lambda$ is a hyperparameter associated with the $\ell_{2,1}$-norm of the matrix $\mathbf{F}$. It plays a role in regulating the balance of clustering results in the objective function. By adjusting the value of $\lambda$, we can control the extent to which the model focuses on the balance of clusters. When the value of $\lambda$ is large, the model will emphasize the balance of samples in each cluster more, making the number of data points in each cluster as close as possible. The selection of $\lambda$ depends on the structural characteristics of the dataset. A larger $\lambda$ is assigned when the dataset has equal sample sizes per class to ensure balance, while a smaller $\lambda$ is used for uneven distributions to balance results while respecting the original structure. The selection of $\lambda$ will impact the performance, demonstrating the regularization term of $\ell_{2,1}$-norm is effective.
>
>
> **2. Centerless strategy may introduce extra computational overhead**
>
> **A**: Our centerless strategy indeed eliminates the dependence on initial centroids, thereby enhancing the robustness of the algorithm. The core of the centerless strategy lies in computing the distances between sample pairs rather than the distances from samples to cluster centers. This requires us to construct a distance matrix $\mathbf{D} \in \mathbb{R}^{n \times n}$, and then build a similarity matrix $\mathbf{S} \in \mathbb{R}^{n \times n}$ through the cluster indicator matrix $\mathbf{F} \in \mathbb{R}^{n \times c}$, where $\mathbf{S} = \mathbf{F}\mathbf{F}^\top$. We initialize and update the cluster indicator matrix $\mathbf{F}$ instead of relying on the initialization and update of centroids. The complexity required to update the cluster indicator matrix $\mathbf{F}$ through formula (36) is $\mathcal{O}(n^2 c)$ for each iteration. Here, we introduce an acceleration strategy by precomputing and storing $\mathbf{F}^\top \mathbf{D}$, which has a time complexity of $\mathcal{O}(n^2 c)$. Then, we iteratively solve for matrix $\mathbf{F}$ with a time complexity of $\mathcal{O}(n c)$. Therefore, the overall computational complexity of updating $\mathbf{F}$ is $\mathcal{O}(n^2 c + t_1 n c)$, $t_1$ represents the number of iterations. Therefore, the centerless strategy truly increases the complexity, but the acceleration strategy helps to alleviate the computational burden. Although our complexity is $\mathcal{O}(n^2)$, the convergence experiment in the appendix shows that our model can converge quickly within 5 iterations.
>
> **3. Initialization of $\mathbf{F}$**
>
> **A**:  We initialize the $\mathbf{F} \in \mathbb{R}^{n \times c}$ by setting an identity matrix to every $c$ rows. Specifically, we take the first $c$ rows as an identity matrix, then the second $c$ rows as an identity matrix, and so on.
>
> **4. Convergence conditions for Algorithm 1 and Algorithm 2**
>
> **A**:  The convergence condition for Algorithm 1 is to check whether the difference between the updated matrix $\mathbf{F}$ and the previous matrix $\mathbf{F}$ is zero. If the difference is zero, the algorithm terminates. For Algorithm 2, we calculate the difference between the objective function values before and after each update of matrix $\mathbf{F}$. If the difference is less than $10^{-6}$, the loop terminates.
>
> **5. The reason of "problem (24) only reaches a maximum when F is a discrete label matrix"**
>
> **A**: Problem (24) is a convex optimization problem because the function ${\sum_{k=1}}\mathbf{F}^{2}\_{ik}$ is convex, and the constraints $0 \leq \mathbf{F}\_{ik} \leq 1$, and $\sum\_{k=1}^c \mathbf{F}\_{ik} = 1$ define a compact set. According to convex optimization theory, if a convex optimization problem has a solution on a compact set, then the solution must lie at one of the extreme points of that set. For the compact set $\lbrace \mathbf{f} \in \mathbb{R}^c | \mathbf{f}^\top \mathbf{1} = 1, \mathbf{f} \geq 0 \rbrace$, its extreme points can only be the $c$ one-hot vectors. Thus, we can conclude that problem (24) only reaches a maximum when $\mathbf{F}$ is a discrete indicator matrix.
>
> **6. Does graph construction introduce additional computational complexity?**
>
> **A**: In Appendix A.4, we provide details on how to construct the input graph for the low-pass filtering distance. The computational complexity for selecting anchor points using the DAS method and constructing the anchor graph is $\mathcal{O}(nrd + nr \log(r))$, where $n$, $r$, and $d$ represent the number of samples, anchor points, and dimensions, respectively. Therefore, the graph construction will only incur a linear complexity with regard to sample number, which is efficient and will not introduce additional computational complexity.

---

> > ### Comment · Reviewer_w5A2 · 2025-04-03
> >
> > The authors have addressed my previous concerns. Therefore, I decide to raise my score to 4.

---

> > > ### Author Response · Authors · 2025-04-06
> > >
> > > Thank you for your support and recognition of our work. We will carefully revise the final version based on your valuable feedback to further improve the quality of the paper. We sincerely appreciate the thoughtful suggestions you have provided and the time and effort you have dedicated to this process.

---

### Decision · Program_Chairs · 2025-05-01

**Decision:**

Accept (poster)

**Comment:**

This paper focuses on three issues in K-means clustering, sensitivity to initial centroid selection, limited ability to discern the intrinsic manifold structures, difficulty in achieving balanced clustering. To solve them, this paper utilizes a label matrix to generate a similarity matrix to align manifold structures with labels, and  integrates Gaussian kernel distance and K-nearest neighbor distance as well as low-pass filtering distance. Further, this paper designs a $\ell_{2,1}$ regularizer to improve the balanced clustering results. Experiments confirm the effectiveness of the proposed algorithm.



After rebuttal, all reviews are positive about this manuscript and recognize its contributions. After reading the manuscript and responses, I agree with reviewers and recommend an acceptance for this manuscript.